# Investigations into the Structure/Antibacterial Activity Relationships of Cyclam and Cyclen Derivatives

**DOI:** 10.3390/antibiotics8040224

**Published:** 2019-11-14

**Authors:** Luis G. Alves, João F. Portel, Sílvia A. Sousa, Olga Ferreira, Stephanie Almada, Elisabete R. Silva, Ana M. Martins, Jorge H. Leitão

**Affiliations:** 1Centro de Química Estrutural, Associação do Instituto Superior Técnico para a Investigação e Desenvolvimento, 1049-003 Lisboa, Portugal; 2IBB-Instituto de Bioengenharia e Biociências, Instituto Superior Técnico, Universidade de Lisboa, 1049-001 Lisboa, Portugalsousasilvia@tecnico.ulisboa.pt (S.A.S.); 3Centro de Química Estrutural, Instituto Superior Técnico, Universidade de Lisboa, 1049-001 Lisboa, Portugalana.martins@tecnico.ulisboa.pt (A.M.M.); 4BioISI—Biosystems & Integrative Sciences Institute, Faculdade de Ciências da Universidade de Lisboa, Campo Grande, 1749-016 Lisboa, Portugal; orferreira@fc.ul.pt (O.F.);; 5CERENA—Centro de Recursos Naturais e Ambiente, Instituto Superior Técnico, Universidade de Lisboa, 1049-001 Lisboa, Portugal

**Keywords:** tetraazamacrocycles, Antibacterials, Azarings, *E. coli*, *S. aureus*

## Abstract

A series of cyclam- and cyclen-derived salts are described in the present work; they were designed specifically to gain insights into their structure and antibacterial activity towards *Staphylococcus aureus* and *Escherichia coli*, used respectively, as Gram-positive and Gram-negative model organisms. The newly synthesized compounds are monosubstituted and *trans*-disubstituted tetraazamacrocycles that display benzyl, methylbenzyl, trifluoromethylbenzyl, or trifluoroethylbenzyl substituents appended on the nitrogen atoms of the macrocyclic ring. The results obtained show that the chemical nature, polarity, and substitution patterns of the benzyl groups, as well as the number of pendant arms, are critical parameters for the antibacterial activity of the cyclam-based salts. The most active compounds against both bacterial strains were the *trans*-disubstituted cyclam salts displaying CF_3_ groups in the *para*-position of the aromatic rings of the macrocyclic pendant arms. The analogous cyclen species presents a lower activity, revealing that the size of the macrocyclic backbone is an important requirement for the antibacterial activity of the tetraazamacrocycles. The nature of the anionic counterparts present on the salts was found to play a minor role in the antibacterial activity.

## 1. Introduction

The discovery and use of antibiotics has revolutionized modern medicine, reaching its golden age between the 1940s and the mid-1960s with the discovery of β-lactams, aminoglycosides, tetracyclines, glycopeptides, macrolides, chloramphenicols, ansamycins, and streptogramins [1], which are still currently in clinical use. In the 1980s and 1990s, many pharmaceutical companies abandoned antibiotics research and development, mainly because of the huge investment required and regulatory barriers [2]. Together with the shortage of investment in novel antibiotics [3], a steady worldwide increase in microbial resistance to antibiotics has been reported. Presently, resistance to multiple antibiotics is estimated to cause a total of 700,000 deaths per year worldwide. This impressive death toll is estimated to reach 10 million by 2050, with a huge negative social and economic global impact. Furthermore, infections caused by multiresistant organisms is presently associated with extended hospitalization periods causing a severe burden to healthcare systems worldwide [4]. A particular group of bacterial pathogens encompassing *Enterococcus faecium*, *Staphylococcus aureus*, *Klebsiella pneumoniae*, *Acinetobacter baumanni*, *Pseudomonas aeruginosa*, and Enterobacter species (the ESKAPE group) has been highlighted as being of particular concern due to their increasing prevalence in hospital infections and resistance to multiple antibiotics [5]. The shortage of clinically effective antimicrobials to treat multiresistant bacteria is driving the search for novel antibiotics with novel modes of action or with interacting targets distinct from those already known for compounds which are clinically in use [6]. 

Cyclam and cyclen are tetraazamacrocycles capable of strongly binding to a wide range of metal ions with numerous applications in medicine [7,8,9,10]. Remarkably, a bis-cyclam derivative was found to be a highly potent and selective inhibitor of HIV; it functions by interacting with receptor number 4 for cytokines that display a Cys–X–Cys sequence (CXCR4), which is used by HIV to enter T-cells [7]. The virus’ access is blocked through the binding of the bis-cyclam amine (-NH) groups with the negatively charged carboxyl groups of aspartate and glutamate residues of the CXCR4 receptor [11]. The detailed understanding of the virus entrance mechanism led to the development of novel cyclam-based antivirals [12]. Tetraazamacrocycles and their metal complexes have also been studied as antimalarial [13,14], antischistosomal [15], as well as antileishmanial drugs [16]. Furthermore, the latter compounds have been investigated as anticancer agents [17]. In this context, a Zn^2+^ tetraazamacrocycle complex exhibiting a considerable cytotoxicity towards human breast, cervical, and lung cancer cell lines was recently described [18]. The interaction with DNA suggests that the mechanism of apoptotic induction may be distinct from that of cisplatin [18]. On the other hand, the antibacterial properties of cyclams and cyclens are scarcely reported, being their activity against *Mycobacterium tuberculosis* the principal focus of interest [19,20]. In a recent work, we have shown that the cyclam salt [H_2_{H_2_(^4-CF^_3_PhCH_2_)_2_Cyclam}](CH_3_COO)_2_.(CH_3_COOH)_2_ has antimicrobial activity against *Escherichia coli*, *Staphylococcus aureus,* and *Pseudomonas aeruginosa*. Furthermore, this compound was also found to be active against selected fungal strains of the *Candida* genus, with modest minimal inhibitory concentration (MIC) values [21]. These results, together with the fact that cyclams and cyclens represent a highly versatile backbone, which is easily modifiable by the introduction of new chemical substituents at the amino groups, highlight these molecules as a potential new family of antimicrobials. 

In the present work, we describe novel cyclam and cyclen derivatives, specifically designed to investigate the relationships between structure and antibacterial activity, with the overall objective of improving the antibacterial properties of these molecules.

## 2. Results and Discussion

The monosubstituted cyclam salt, **5**, was synthesized according to a procedure that involves the protection of three of the four nitrogen atoms of the cyclam ring in order to selectively functionalize only one of them. As so, protection was done using *tert*-butyloxycarbonyl (Boc) groups through the reaction of cyclam, **1**, with three equiv. of Boc_2_O (see Scheme 1). This procedure was already reported in the literature to give the trisubstituted derivative H(Boc)_3_Cyclam, **2** [22]. The alkylation of **2** with 4-(trifluoromethyl)benzyl bromide led to the formation of (^4-CF3^PhCH_2_)(Boc)_3_Cyclam, **3**, which was converted into the chloride salt **5** after removal of all Boc protecting groups. Two pathways for the synthesis of **5** were explored: i) The reaction of **3** with trifluoracetic followed by neutralization with KOH to give the neutral compound **4**, which was subsequently protonated with HCl; and ii) the direct reaction of **3** with HCl. The synthetic route for the preparation of [H_4_{H_2_(^4-CF3^PhCH_2_)Cyclam}]Cl_4_, **5**, is presented in Scheme 1. The two procedures led essentially to the same overall yield of **5**, which shows that the two-step path is not necessary if the cyclam salt is the desired product. 

The ^1^H NMR spectrum of **5** reveals ten multiplets corresponding to the methylene protons of the macrocycle backbone integrating to two protons each, and one singlet that corresponds to the two benzylic protons of the pendant arm as a result of the C_1_ symmetry of the compound. In addition, two doublets integrating to two protons each appear in the aromatic region of the spectrum. The NH_2_^+^ protons are not observed in the D_2_O solution, in agreement with proton exchange. The ^13^C{^1^H} NMR spectrum of **5** displays ten different resonances for the macrocycle carbons and one signal for the benzylic carbon of the pendant arm, as expected. Additional resonances due to aromatic rings and the CF_3_ group are also present. Despite the overlapping of several resonances observed in the proton and carbon NMR spectra, their assignment was based on 2D NMR experiments (^1^H-^1^H COSY and ^1^H-^13^C HSQC). The ^19^F NMR spectrum shows a singlet due to the CF_3_ group at −62.6 ppm. 

The preparation of *trans*-disubstituted cyclams of general formula H_2_Bn_2_Cyclam (Bn = PhCH_2_, **6**, ^4-CF3^PhCH_2_, **7**, ^4-CH3^PhCH_2_, **8**, ^4-CF3CH2^PhCH_2_, **9**, and ^3-CF3^PhCH2, **10**) was attained through a standard procedure that involves the reaction of 1,4,8,11-tetraazatriciclo[9.3.1.14,8]hexadecane with 2 equiv. of the suitable benzyl bromide, followed by basic hydrolysis [23,24,25].

In the ^1^H NMR spectra of compounds **6**–**10**, the macrocycle geminal protons are equivalent, giving rise to the emergence of only five signals integrating to four protons each. The methylene protons of the pendant arms show up as singlets in accordance with the fast nitrogen inversion that determines C_2_ symmetry in the solution. The ^13^C{^1^H} NMR spectra display five different resonances for the macrocycle backbone and one set of resonances that corresponds to the benzyl moieties. The proton and carbon NMR spectra of **6**–**10** are similar to those obtained for other *trans*-disubstituted cyclams already reported and do not deserve further comment [23,24,25].

Crystals of **8** suitable for single crystal X-ray diffraction were obtained from slow evaporation of a chloroform solution. Crystallographic and experimental details of data collection and crystal structure determination are presented in the experimental section. A Mercury diagram of the solid-state molecular structure of **8** is shown in Figure 1. 

The solid-state molecular structure of **8** shows the two benzyl pendant arms located at opposite sides of the macrocyclic ring. Despite the structural arrangement of the cyclam ring, one cannot consider intramolecular hydrogen bonds between N(2)-H(2N) and N(1) as the corresponding angles are narrower than 110° [26]. These features were also observed in the previously reported solid-state molecular structure of H_2_(^4-CN^PhCH_2_)_2_Cyclam [25].

Compounds **6**–**10** were converted into the corresponding acetate salts (**11**–**15**) in high yields upon protonation of the two remaining NH groups of the macrocycles with acetic acid. The chloride and bromide salts (**16**–**21**) were obtained by the addition of concentrated aqueous solutions of HCl and HBr to ethanolic solutions of compounds **6**–**10**, respectively. The latter compounds precipitate out of the solution in very high yields. The synthetic route for the preparation of compounds **6**–**21** is shown in Scheme 2.

The ^1^H NMR spectra of compounds **11**–**21** are similar to the ones described for the parent species **6**–**10** showing five multiplets corresponding to the methylene protons of the macrocycle backbone integrating to four protons each and one singlet that correspond to the four benzylic protons of the pendant arms. In addition, one set of resonances appears in the aromatic region of the spectra. In compounds **13** and **18**, the protons of the CH_3_ groups show up as singlets at 2.32 and 1.37 ppm, respectively. In **14** and **19**, the CH_2_CF_3_ groups appear as a quartet with ^3^*J*_H-F_ = 11 Hz at 3.47 and 3.61 ppm, respectively. The NH_2_^+^ and COOH protons are absent in D_2_O solutions due to fast proton exchange. The ^13^C{^1^H} NMR spectra of **11**–**21** are also similar to the ones described for **6**–**10** displaying five different resonances for the macrocycle and one for the benzylic carbon of the pendant arms, as expected. Additional resonances due to aromatic rings are also present. 

Crystals of **14** suitable for single crystal X-ray diffraction were obtained by the concentration of an aqueous acetic acid solution. The asymmetric unit of **14** displays two molecules (**14a** and **14b**). Crystallographic and experimental details of the data collection and crystal structure determination are presented in the experimental section. The solid-state molecular structure of **14b** is presented in Figure 2. It shows two mutually *trans*-benzyl pendant arms pointing to opposite sides of the macrocyclic ring. The structural arrangement of **14** reveals the establishment of hydrogen bonds between the acetate anions, the cyclam framework, and co-crystallized acetic acid molecules. Detailed hydrogen bond lengths and angles are presented in Appendix A. Each [CH_3_COO…HOOCCH_3_]^−^ pair is located on opposite sides of the plane that contains the four nitrogen atoms of the cyclam ring. Such interactions are not present in the solution as revealed by the C_2_ symmetry observed in the NMR spectra. The solid-state molecular structure of **14** reveals similar features to those observed for [H_2_{H_2_(^4-CF3^PhCH_2_)_2_Cyclam}](CH_3_COO)_2_·(CH_3_COOH)_2_, **12**, as previously reported [21].

The protonation of cyclam with HCl leads to the production of chloride salt [H_4_(H_4_Cyclam)]Cl_4_, **22**, in a very high yield (see Scheme 3). 

The NMR spectra of **22** are consistent with a tetraprotonated cyclam salt revealing only three resonances in both the proton and carbon spectra that correspond to the CH_2_ groups of the [C2] and [C3] chains of the cyclam ring.

The *trans*-disubstituted cyclen **24** was prepared by reaction of the bisaminal cyclen derivative **23** with 2 equiv. of ^4-CF3^PhCH_2_Br, followed by hydrazinolysis using hydrazine monohydrate. The corresponding chloride salt **25** was obtained by protonation of the neutral cyclen species **24** with a concentrated aqueous solution of HCl, as presented in Scheme 4.

The NMR spectra of **24** and **25** are similar showing the typical pattern for a *trans*-disubstituted cyclen. The ^1^H NMR spectra show two multiplets corresponding to the methylene protons of the macrocycle backbone integrating to eight protons each and one singlet that correspond to the four benzylic protons of the pendant arms. In addition, two doublets appear in the aromatic region of the spectra. The ^13^C{^1^H} NMR spectra of both compounds display two different resonances for the macrocycle and one for the benzylic carbon of the pendant arms, as expected. Additional resonances due to the aromatic ring carbons and the CF_3_ group are also present. The ^19^F NMR spectra of **24** and **25** show a singlet due to the CF_3_ groups at −62.5 and −59.8 ppm, respectively.

The structure/antibacterial activity relationships of cyclam salts **5**, **11**–**22**, and **25** were assessed based on the determination of the MIC values of the compounds towards *E. coli* ATCC 25922 and *S. aureus* Newman. The results obtained for both bacteria are presented in Table 1.

The results presented in Table 1 show that in general the counter anion did not significantly affect the antibacterial activity of the compounds, and presented similar MIC values, as evidenced by sets **14**/**19**, **15**/**20**, and **12**/**17**/**21**. The minor variations observed most probably result from the different molecular mass of acetate and chloride. This effect is more evident on the pair **17**/**21**, with **21** exhibiting slightly higher MIC values most probably because of the higher molecular mass of bromine. An exception to these general observations was detected for the pairs **11**/**16** and **13**/**18**. In those cases, the cyclam derivative has apolar pendant arms and the discrepancies observed for the MIC values of the acetate and chloride salts might result from additional unknown factors dependent on the strain under study. Such observations are more pronounced for *S. aureus* than for *E. coli*, and might reflect the distinct and specific mechanisms used by each bacterial species to cope with antimicrobials. 

The results in Table 1 also evidence the effect of the number of pendant arms on the antibacterial activity of cyclam derivatives: MIC values higher than 512 µg/mL were estimated for the monosubstituted cyclam **5**, while MIC values of 7.3 and 4.3 µg/mL were registered for the disubstituted cyclam derivative **17** towards *S. aureus* and *E. coli*, respectively.

We also investigated the effect of the distance of the trifluoromethyl group attached to the aromatic ring of the macrocyclic pendant arm on the antibacterial activity of the compound. The results in Table 1 show that the insertion of one CH_2_ spacer between the trifluoromethyl group and the aromatic ring led to a slight reduction in the antibacterial activity of cyclam derivatives (see pairs **12**/**14** and **17**/**19**). The decrease in antimicrobial activity may have either an electronic or stereochemical origin. The inclusion of a CH_2_ spacer has a strong influence on the relative position of the CF_3_ group towards a possible acceptor fragment, and thus, it is expected to modify the interaction between both fragments. Additionally, the CH_2_ spacer blocks the electronic delocalization that is present if the CF_3_ group is directly bonded to the aromatic ring. The importance of the substitution pattern of the phenyl ring is also attested by the comparison of antibacterial properties of the *meta*- and *para*-CF_3_ cyclam pending groups (see pairs **12**/**15** and **17**/**20**). One may thus conclude that the antibacterial activity is ruled by subtle electronic interactions between the molecule and the receptor [27,28]. In line with these considerations, the results in Table 1 show that the polarity of the substituent on the aromatic ring of the macrocyclic pendant arm is critical for the antimicrobial activity of the compounds. Cyclam derivatives with the polar CF_3_ substituents present higher antibacterial activities than those with the apolar CH_3_ substituents (see MIC values of pairs **12**/**13** and **17**/**18**). 

The analogous cyclen species **25** presents a lower antimicrobial activity. This result suggests that the size of the macrocyclic backbone is an important feature for the antimicrobial activity of the tetraazamacrocycles, although strong conclusions cannot be taken on this subject due to the limited number of compounds tested.

## 3. Materials and Methods

### 3.1. General Considerations

Compounds **1** [29], **2** [22], **6** [23], **7** [24], **12** [21], **23** [30], and 1,4,8,11-tetraazatriciclo[9.3.1.14,8]hexadecane [23] were prepared according to published procedures. All other reagents were commercial grade and used without further purification. NMR spectra were recorded in a Bruker AVANCE II 300 or 400 MHz spectrometer, at 296 K unless stated otherwise, referenced internally to residual proton-solvent (^1^H) or solvent (^13^C) resonances, and reported relative to tetramethyl silane (0 ppm). ^19^F NMR was referenced to external CF_3_COOH (−76.55 ppm). ^1^H-^13^C{^1^H} HSQC and ^1^H-^1^H COSY NMR experiments were performed in order to perform all the assignments. Elemental analyses were obtained from Laboratório de Análises do IST.

### 3.2. Synthetic Procedures

Boc_3_(^4-CF3^PhCH_2_)Cyclam (**3**): Compound **2** (1.02 g, 2.04 mmol) was dissolved in a minimum volume of dimethylformamide. K_2_CO_3_ (0.70 g, 5.06 mmol) and 4-(trifluoromethyl)benzyl bromide (0.49 g, 2.05 mmol) were added. The reaction mixture was left stirring overnight. A saturated solution of KHCO_3_ and brine were added, and the product extracted with small portions of chloroform. The organic phases were combined and dried with anhydrous MgSO_4_. After filtration, the solvent was evaporated under reduced pressure and the product was obtained in a 98% yield (1.32 g, 2.00 mmol). ^1^H NMR (CDCl_3_, 300.1 MHz, 296 K): δ (ppm) 7.50 (d, ^3^*J*_H-H_ = 8 Hz, 2H, *o-Ph* or *m-Ph*), 7.34 (d, ^3^*J*_H-H_ = 8 Hz, 2H, *o-Ph* or *m-Ph*), 3.53 (s, 2H, PhC*H*_2_N), 3.30 (overlapping, 12H total, 6H, [C3]*C*H_2_N and 6H, [C2]*C*H_2_N), 2.56 (m, 2H [C2]*C*H_2_N), 2.34 (m, 2H [C3]*C*H_2_N), 1.84 (m, 2H, CH_2_C*H*_2_CH_2_), 1.65 (m, 2H, CH_2_C*H*_2_CH_2_), 1.41-1.21 (br, 27H, C*H*_3_). ^13^C{^1^H} NMR (CDCl_3_, 75.5 MHz, 296 K): δ (ppm) 155.8 (*C*O), 155.6 (overlapping, *C*O), 143.4 (*i-Ph*), 129.2 (*o-Ph* or *m-Ph*), 127.4 (q, ^2^*J*_C-F_ = 97 Hz, *p-Ph*), 125.2 (*o-Ph* or *m-Ph*), 124.3 (q, ^1^*J*_C-F_ = 272 Hz, *C*F_3_), 79.7 (*C*(CH_3_)_3_), 79.6 (*C*(CH_3_)_3_), 79.5 (*C*(CH_3_)_3_), 59.5 (Ph*C*H_2_N), 53.5 ([C2]*C*H_2_N), 51.9 ([C3]*C*H_2_N), 47.9-46.3 (overlapping, [C2]*C*H_2_N and [C3]*C*H_2_N), 28.5-28.4 (overlapping, CH_2_*C*H_2_CH_2_ and *C*H_3_). ^19^F NMR (CDCl_3_, 282.4 MHz, 296K): δ (ppm) −62.4 (s, C*F*_3_). MS (CH_3_CN, ESI): *m/z* 659.21 [M+H]^+^, 501.20 [M-^CF3^PHCH_2_+H]^+^.

H_3_(^4-CF3^PhCH_2_)Cyclam (**4**): Compound **3** (0.66 g, 1.00 mmol) was dissolved in 20 mL of dichloromethane and 10 mL of trifluoracetic acid (0.13 mol) were added. The reaction mixture was refluxed overnight. The solvent was evaporated to dryness to give a brow oil that was dissolved in water. KOH was added until the reaction mixture reached pH ≈ 13. The product was extracted with dichloromethane, the organic phase was washed with brine and dried with anhydrous MgSO_4_. After filtration, the solvent was evaporated to dryness producing the product as a white solid in a 31% yield (0.11 g, 0.31 mmol). ^1^H NMR (CDCl_3_, 400.1 MHz, 296 K): δ (ppm) 7.57 (d, ^3^*J*_H-H_ = 8 Hz, 2H, *m-Ph*), 7.48 (d, ^3^*J*_H-H_ = 8 Hz, 2H, *o-Ph*), 4.39 (br, 3H, N*H*) 3.65 (s, 2H, PhC*H*_2_N), 2.90 (m, 2H, [C2]*C*H_2_N), 2.85 (m, 2H [C3]*C*H_2_N), 2.79 (overlapping, 4H total, 2H, [C3]*C*H_2_N and 2H, [C2]*C*H_2_N), 2.71 (overlapping, 4H total, 2H, [C3]*C*H_2_N and 2H, [C2]*C*H_2_N), 2.60 (overlapping, 4H total, 2H, [C3]*C*H_2_N and 2H, [C2]*C*H_2_N), 1.89 (m, 2H, CH_2_C*H*_2_CH_2_), 1.81 (m, 2H, CH_2_C*H*_2_CH_2_). ^13^C{^1^H} NMR (CDCl_3_, 100.6 MHz, 296 K): δ (ppm) 142.8 (*i-Ph)*, 129.7 (q, ^2^*J*_C-F_ = 30 Hz, *p-Ph*), 129.6 (*o-Ph*), 127.0 (q, ^1^*J*_C-F_ = 264 Hz, *C*F_3_), 125.5 (q, ^3^*J*_C-F_ = 4 Hz, *m-Ph*), 57.9 (Ph*C*H_2_N), 53.5 ([C2]*C*H_2_N or [C3]*C*H_2_N), 53.1 [C2]*C*H_2_N or [C3]*C*H_2_N), 50.5 ([C2]*C*H_2_N or [C3]*C*H_2_N), 49.3 ([C3]*C*H_2_N), 48.8 ([C2]*C*H_2_N or [C3]*C*H_2_N), 48.6 ([C2]*C*H_2_N or [C3]*C*H_2_N), 47.2 ([C2]*C*H_2_N or [C3]*C*H_2_N), 47.1 ([C2]*C*H_2_N), 27.0 (CH_2_*C*H_2_CH_2_), 25.6 (CH_2_*C*H_2_CH_2_). ^19^F NMR (CDCl_3_, 376.5 MHz, 296K): δ (ppm) −62.4 (s, C*F*_3_). Anal. calcd for C_18_H_29_F_3_N_4_: C, 60.31; H, 8.16; N, 15.63. Found: C, 60.28; H, 8.19; N, 15.57.

[H_4_{H_3_(^4-CF3^PhCH_2_)Cyclam}]Cl_4_ (**5**): ***Method A***: Compound **3** (0.66 g, 1.00 mmol) was dissolved in dichloromethane and a concentrated aqueous solution of HCl (37%) was added until pH ≈ 1. The reaction mixture was left stirring overnight at room temperature. The solvent was evaporated to dryness producing the product as white solid in 56% yield (0.29 g, 0.56 mmol). ***Method B***: Compound **4** (0.11 g, 0.31 mmol) was dissolved in ethanol and a concentrated aqueous solution of HCl (37%) was added until pH ≈ 1. The product was precipitated out of the solution. After filtration, the product was dried producing the compound as a white solid in a 48% yield (0.08 g, 0.15 mmol). ^1^H NMR (D_2_O/(CD_3_)_2_CO, 300.1 MHz, 296 K): δ (ppm) 7.87 (d, ^3^*J*_H-H_ = 9 Hz, 2H, *m-Ph*), 7.78 (d, ^3^*J*_H-H_ = 9 Hz, 2H, *o-Ph*), 4.48 (s, 2H, PhC*H*_2_N), 3.70 (m, 6H, [C2]*C*H_2_N), 3.59 (m, 2H [C2]*C*H_2_N), 3.50 (m, 6H, [C3]*C*H_2_N), 3.36 (m, 2H [C3]*C*H_2_N), 2.33 (m, 4H, CH_2_C*H*_2_CH_2_). ^13^C {^1^H} NMR (D_2_O/(CD_3_)_2_CO, 75.5 MHz, 296 K): δ (ppm) 135.1 (*i-Ph)*, 131.8 (*o-Ph*), 131.2 (q, ^2^*J*_C-F_ = 32 Hz, *p-Ph*), 126.4 (d, ^3^*J*_C-F_ = 3 Hz, *m-Ph*), 124.2 (q, ^1^*J*_C-F_ = 272 Hz, *C*F_3_), 57.4 (Ph*C*H_2_N), 48.7 ([C3]*C*H_2_N), 46.3 ([C2]*C*H_2_N), 43.0 ([C3]*C*H_2_N), 42.5 (overlapping, [C3]*C*H_2_N), 39.5 (overlapping, [C2]*C*H_2_N), 39.1 ([C2]*C*H_2_N), 20.0 (CH_2_*C*H_2_CH_2_), 19.7 (CH_2_*C*H_2_CH_2_). ^19^F NMR (D_2_O/(CD_3_)_2_CO, 282.4 MHz, 296K): δ (ppm) −62.6 (s, C*F*_3_). Anal. calcd for C_18_H_33_Cl_4_F_3_N_4_.H_2_O: C, 41.39; H, 6.75; N, 10.73. Found: C, 41.29; H, 6.60; N, 10.68.

H_2_(^4-CH3^PhCH_2_)_2_Cyclam (**8**): 1,4,8,11-tetraazatriciclo[9.3.1.14,8]hexadecane (5.00 g, 22.3 mmol) was dissolved in a minimum volume of acetonitrile and 2.2 equiv. of 4-(methyl)benzyl bromide (9.10 g, 49.2 mmol) were added. The solution was stirred overnight at room temperature resulting in a white precipitate that was filtered, washed with acetonitrile, and dried under reduced pressure. A portion of the product obtained was hydrolyzed in an aqueous NaOH solution (3M) for 4 h under stirring at room temperature. The product was extracted with small portions of chloroform that were combined and dried with anhydrous MgSO_4_. After filtration, the solvent was evaporated to dryness giving an oil that was converted into a white solid after successive freeze–trituration–pump–thaw cycles. Compound **8** was obtained in a 92% yield (3.80 g, 9.30 mmol). Suitable crystals for single crystal X-ray diffraction were obtained from a concentrated chloroform solution. ^1^H NMR (CDCl_3_, 400.1 MHz, 296 K): δ (ppm) 7.19 (d, ^3^*J*_H-H_ = 8 Hz, 4H, *o-Ph* or *m-Ph*), 7.10 (d, ^3^*J*_H-H_ = 8 Hz, 4H, *o-Ph* or *m-Ph*), 3.69 (s, 4H, PhC*H*_2_N), 2.79 (s, 2H, N*H*), 2.73 (m, 4H, [C3]C*H*_2_N), 2.69 (m, 4H, [C2]C*H*_2_N), 2.55 (m, 4H, [C2]C*H*_2_N), 2.51 (m, 4H, [C3]C*H*_2_N), 2.29 (s, 6H, C*H*_3_), 1.83 (m, 4H, CH_2_C*H*_2_CH_2_). ^13^C {^1^H} NMR (CDCl_3_, 100.6 MHz, 296 K): δ (ppm) 136.6 (*i-Ph* or *p-Ph*), 134.2 (*i-Ph* or *p-Ph*), 129.7 (*o-Ph* or *m-Ph*), 128.9 (*o-Ph* or *m-Ph*), 57.5 (Ph*C*H_2_N), 54.2 ([C2]*C*H_2_N), 51.8 ([C3]*C*H_2_N), 50.5 ([C3]*C*H_2_N), 47.9 ([C2]*C*H_2_N), 26.1 (CH_2_*C*H_2_CH_2_), 21.8 (*C*H_3_). Anal. calcd for C_26_H_40_N_4_: C, 76.42; H, 9,87; N, 13.71. Found: C, 76.06; H, 9.91; N, 13.69.

H_2_(^4-CF3CH2^PhCH_2_)_2_Cyclam (**9**): Compound **9** was prepared by the same procedure described for **8** by reaction of 1,4,8,11-tetraazatriciclo[9.3.1.14,8]hexadecane with 1-(bromomethyl)-4-(2,2,2-trifluoroethyl)benzene being obtained as a white solid in a 52% yield (0.28 g, 0.51 mmol). ^1^H NMR (CDCl_3_, 400.1 MHz, 296 K): δ (ppm) 7.29 (d, ^3^*J*_H-H_ = 8 Hz, 4H, *o-Ph* or *m-Ph*), 7.21 (d, ^3^*J*_H-H_ = 8 Hz, 4H, *o-Ph* or *m-Ph*), 3.71 (s, 4H, PhC*H*_2_N), 3.31 (q, 4H, ^3^*J*_H-F_ = 11 Hz, C*H*_2_CF_3_), 2.72-2.53 (overlapping, 18H total, 2H, N*H*, 8H, [C3]C*H*_2_N and 8H, [C2]C*H*_2_N), 1.83 (m, 4H, CH_2_C*H*_2_CH_2_). ^13^C{^1^H} NMR (CDCl_3_, 100.6 MHz, 296 K): δ (ppm) 137.6 (*i-Ph*), 130.1 (*o-Ph* or *m-Ph*), 129.9 (*o-Ph* or *m-Ph*), 129.0 (*p-Ph*), 125.9 (q, ^1^*J*_C-F_ = 279 Hz, CH_2_*C*F_3_), 57.6 (Ph*C*H_2_N), 54.3 ([C2]*C*H_2_N), 51.7 ([C3]*C*H_2_N), 50.1 ([C2]*C*H_2_N or [C3]*C*H_2_N), 47.9 ([C2]*C*H_2_N or [C3]*C*H_2_N), 40.0 (q, ^2^*J*_H-F_ = 29 Hz, *C*H_2_CF_3_), 26.1 (CH_2_*C*H_2_CH_2_). ^19^F NMR (CDCl_3_), 376.5 MHz, 296K): δ (ppm) −65.9 (s, C*F*_3_). Anal. calcd for C_28_H_38_F_6_N_4_: C, 61.75; H, 7.03; N, 10.29. Found: C, 60.82; H, 6.75; N, 10.00.

H_2_(^3-CF3^PhCH_2_)_2_Cyclam (**10**): Compound **10** was prepared by the same procedure described for **8** by reaction of 1,4,8,11-tetraazatriciclo[9.3.1.14,8]hexadecane with 1-(bromomethyl)-3-(trifluoromethyl)benzene being obtained as a white solid in a 61% yield (2.55 g, 4.94 mmol). ^1^H NMR (CDCl_3_, 400.1 MHz, 296 K): δ (ppm) 7.55 (s, 2H, *o-Ph*), 7.49-7.44 (overlapping, 4H total, *o-Ph* and *m-Ph*), 7.37 (d, ^3^*J*_H-H_ = 8 Hz, 2H, *p-Ph*), 3.68 (s, 4H, PhC*H*_2_N), 2.74-2.69 (overlapping, 10H total, 2H, N*H*, 4H, [C3]C*H*_2_N and 4H, [C2]C*H*_2_N), 2.59 (m, 4H, [C2]C*H*_2_N), 2.51 (m, 4H, [C3]C*H*_2_N), 1.83 (m, 4H, CH_2_C*H*_2_CH_2_). ^13^C{^1^H} NMR (CDCl_3_, 100.6 MHz, 296 K): δ (ppm) 139.3 (*i-Ph*), 132.6 (*o-Ph* or *m-Ph*), 130.6 (q, ^2^*J*_C-F_ = 32 Hz, *m-Ph*), 128.7 (*o-Ph* or *m-Ph*), 125.8 (q, ^3^*J*_C-F_ = 3 Hz, *o-Ph*), 124.3, (q, ^1^*J*_C-F_ = 273 Hz, *C*F_3_), 124.0 (q, ^3^*J*_C-F_ = 4 Hz, *p-Ph*), 57.6 (Ph*C*H_2_N), 53.9 ([C2]*C*H_2_N), 51.3 ([C3]*C*H_2_N), 49.1 ([C2]*C*H_2_N or [C3]*C*H_2_N), 47.5 ([C2]*C*H_2_N or [C3]*C*H_2_N), 25.9 (CH_2_*C*H_2_CH_2_). ^19^F NMR (CDCl_3_), 376.5 MHz, 296K): δ (ppm) −62.5 (s, C*F*_3_). Anal. calcd for C_26_H_34_F_6_N_4_: C, 60.45; H, 6.63; N, 10.85. Found: C, 60.21; H, 6.63; N, 10.82.

[H_2_{H_2_(PhCH_2_)_2_Cyclam}](CH_3_COO)_2_.(CH_3_COOH)_2_ (**11**): Compound **6** (0.35 g, 0.92 mmol) was dissolved in a small volume of acetonitrile and 1 mL of glacial acetic acid was added to the solution. This mixture was refluxed for 1 h and the solvent was evaporated under reduced pressure producing a white solid that was washed with diethyl ether and dried in a vacuum. The product was obtained as a white solid in an 84% yield (0.48 g, 0.77 mmol). ^1^H NMR (CDCl_3_, 400.1 MHz, 296 K): δ (ppm) 10.81 (overlapping, 6H total, 4H, N*H*_2_^+^ and 2H, CH_3_COO*H*), 7.31-7.30 (overlapping, 6H total, 2H, *p-Ph* and 4H, *m-Ph*), 7.21 (d, ^3^*J*_H-H_ = 8 Hz, 4H, *o-Ph*), 3.97 (s, 4H, PhC*H*_2_N), 3.25 (m, 4H, [C3]C*H*_2_N), 3.20 (m, 4H, [C2]C*H*_2_N), 2.78 (m, 4H, [C2]C*H*_2_N), 2.74 (m, 4H, [C3]C*H*_2_N), 2.00-1.97 (overlapping, 16H total, 4H, CH_2_C*H*_2_CH_2_, 6H, C*H*_3_COO^−^ and 6H, C*H*_3_COOH). ^13^C{^1^H} NMR (CDCl_3_,100.6 MHz, 296 K): δ (ppm) 176.5 (*C*OO), 134.0 (*i-Ph*), 130.8 (*o-Ph*), 128.4 (*m-Ph*), 127.7 (*p-Ph*), 53.9 (Ph*C*H_2_N), 51.9 ([C3]*C*H_2_N), 48.5 (overlapping, [C2]*C*H_2_N and [C3]*C*H_2_N), 45.4 ([C2]*C*H_2_N), 22.8 (CH_2_*C*H_2_CH_2_), 22.5 (*C*H_3_COO). Anal. calcd for C_28_H_44_N_4_O_4_.(CH_3_COOH)_2_: C, 61.91; N, 9.03; H, 8.44. Found: C, 61.85; N, 9.81; H, 8.40.

[H_2_{H_2_(^4-CH3^PhCH_2_)_2_Cyclam}](CH_3_COO)_2_.(CH_3_COOH)_2_ (**13**): Compound **13** was prepared by the same procedure described for **11** using compound **8** (1.00 g, 2.45 mmol) as the starting material. The product was obtained as a white solid in an 80% yield (1.27 g, 1.96 mmol). ^1^H NMR (D_2_O, 300.1 MHz, 296 K): δ (ppm) 7.31 (d, ^3^*J*_H-H_ = 6 Hz, 4H, *o-Ph* or *m-Ph*), 7.23 (d, ^3^*J*_H-H_ = 6 Hz, 4H, *o-Ph* or *m-Ph*), 3.48 (s, 4H, PhC*H*_2_N), 3.31 (m, 4H, [C3]C*H*_2_N or [C2]C*H*_2_N), 3.23 (m, 4H, [C3]C*H*_2_N or [C2]C*H*_2_N), 2.80 (m, 4H, [C3]C*H*_2_N or [C2]C*H*_2_N), 2.76 (m, 4H, [C3]C*H*_2_N or [C2]C*H*_2_N), 2.25 (s, 6H, C*H*_3_), 1.99 (overlapping, 16H total, 4H, CH_2_C*H*_2_CH_2_, 6H, C*H*_3_COO^-^ and 6H, C*H*_3_COOH). ^1^H NMR (CDCl_3_, 300.1 MHz, 296 K): δ (ppm) 10.73 (overlapping, 6H total, 4H, N*H*_2_^+^ and 2H, CH_3_COO*H*), 7.12 (d, ^3^*J*_H-H_ = 6 Hz, 4H, *o-Ph* or *m-Ph*), 7.02 (d, ^3^*J*_H-H_ = 9 Hz, 4H, *o-Ph* or *m-Ph*), 3.79 (s, 4H, PhC*H*_2_N), 3.16-3.14 (overlapping, 8H total, 4H, [C3]C*H*_2_N and 4H, [C2]C*H*_2_N), 2.75 (m, 4H, [C2]C*H*_2_N), 2.66 (m, 4H, [C3]C*H*_2_N), 2.32 (s, 6H, C*H*_3_), 1.99 (overlapping, 16H total, 4H, CH_2_C*H*_2_CH_2_, 6H, C*H*_3_COO^-^ and 6H, C*H*_3_COOH). ^13^C{^1^H} NMR (CDCl_3_, 75.5 MHz, 296 K): δ (ppm) 176.6 (*C*OO), 137.3 (*i-Ph*), 130.9 (*p-Ph*), 130.8 (*o-Ph* or *m-Ph*), 129.1 (*o-Ph* or *m-Ph*), 52.7 (Ph*C*H_2_N), 51.3 ([C3]*C*H_2_N), 48.7 ([C2]*C*H_2_N), 48.3 ([C3]*C*H_2_N), 45.6 ([C2]*C*H_2_N), 22.8 (CH_2_*C*H_2_CH_2_), 22.6 (*C*H_3_COO), 21.2 (*C*H_3_). Anal. calcd for C_30_H_48_N_4_O_4_.(CH_3_COOH)_2_: C, 62.94; H,8.70; N,8.64. Found: C, 62.26; H, 9.17; N, 8.46.

[H_2_{H_2_(^4-CF3CH2^PhCH_2_)_2_Cyclam}](CH_3_COO)_2_.(CH_3_COOH)_2_ (**14**): Compound **14** was prepared by the same procedure described for **11** using compound **9** (0.24 g, 0.44 mmol) as the starting material. The product was obtained as a white solid in an 84% yield (0.29 g, 0.37 mmol). Suitable crystals for single crystal X-ray diffraction were obtained from an aqueous acetic acid solution. ^1^H NMR (D_2_O, 400.1 MHz, 296 K): δ (ppm) 7.46 (d, ^3^*J*_H-H_ = 8 Hz, 4H, *o-Ph* or *m-Ph*), 7.35 (d, ^3^*J*_H-H_ = 8 Hz, 4H, *o-Ph* or *m-Ph*), 3.60 (s, 4H, PhC*H*_2_N), 3.47 (q, 4H, ^3^*J*_H-F_ = 11 Hz, C*H*_2_CF_3_), 3.33 ([C3]C*H*_2_N or [C2]C*H*_2_N), 3.28 (m, 4H, [C3]C*H*_2_N or [C2]C*H*_2_N), 2.83 (m, 4H, [C3]C*H*_2_N or [C2]C*H*_2_N), 2.73 (m, 4H, [C3]C*H*_2_N or [C2]C*H*_2_N), 1.99 (overlapping, 16H total, 4H, CH_2_C*H*_2_CH_2_, 6H, C*H*_3_COO^-^ and 6H, C*H*_3_COOH). ^1^H NMR (CDCl_3_, 400.1 MHz, 296 K): δ (ppm) 10.28 (overlapping, 6H total, 4H, N*H*_2_^+^ and 2H, COO*H*), 7.25 (d, ^3^*J*_H-H_ = 8 Hz, 4H, *o-Ph* or *m-Ph*), 7.16 (d, ^3^*J*_H-H_ = 8 Hz, 4H, *o-Ph* or *m-Ph*), 3.83 (s, 4H, PhC*H*_2_N), 3.34 (m, ^3^*J*_H-F_ = 11 Hz, 4H, C*H*_2_CF_3_), 3.10 (overlapping, 8H total, 4H, [C2]C*H*_2_N and 4H, [C3]C*H*_2_N), 2.76 (m, 4H, [C2]C*H*_2_N), 2.68 (m, 4H, [C3]C*H*_2_N), 2.00-1.96 (overlapping, 16H total, 4H, CH_2_C*H*_2_CH_2_, 6H, C*H*_3_COO^-^ and 6H, C*H*_3_COOH). ^13^C{^1^H} NMR (CDCl_3_,100.6 MHz, 296 K): δ (ppm) 176.7 (*C*OO), 134.6 (*i-Ph*), 131.0 (*o-Ph* or *m-Ph*), 130.2 (*o-Ph* or *m-Ph*), 129.6 (m, ^3^*J*_C-F_ = 3 Hz, *p-Ph*), 125.8 (q, ^1^*J*_C-F_ = 277 Hz, CH_2_*C*F_3_), 53.3 (Ph*C*H_2_N), 51.1 ([C3]*C*H_2_N), 49.1 ([C2]*C*H_2_N), 48.2 ([C3]*C*H_2_N or [C2]*C*H_2_N), 45.9 ([C3]*C*H_2_N or [C2]*C*H_2_N), 40.0 (q, ^2^*J*_H-F_ = 29 Hz, *C*H_2_CF_3_), 23.1 (CH_2_*C*H_2_CH_2_), 22.6 (*C*H_3_COO). ^19^F NMR (D_2_O, 376.5 MHz, 296 K): δ (ppm) −65.8 (s, CH_2_C*F*_3_). ^19^F NMR (CDCl_3_, 376.5 MHz, 296 K): δ (ppm) −65.9 (s, CH_2_C*F*_3_). Anal. calcd for C_32_H_46_F_6_N_4_O_4_.(CH_3_COOH)_2_: C, 55.09; H, 6.94; N, 7.14. Found: C, 55.00; H, 6.86; N, 7.01.

[H_2_{H_2_(^3-CF3^PhCH_2_)_2_Cyclam}](CH_3_COO)_2_.(CH_3_COOH)_2_ (**15**): Compound **15** was prepared by the same procedure described for **11** using compound **10** (0.55 g, 1.06 mmol) as the starting material. The product was obtained as a white solid in a 28% yield (0.23 g, 0.30 mmol). ^1^H NMR (D_2_O, 400.1 MHz, 296 K): δ (ppm) 7.70 (d, ^3^*J*_H-H_ = 8 Hz, 2H, *p-Ph*), 7.65 (s, 2H, *o-Ph*), 7.62 (t, ^3^*J*_H-H_ = 8 Hz, 2H, *m-Ph*), 7.56 (d, ^3^*J*_H-H_ = 8 Hz, 2H, *o-Ph*), 3.70 (s, 4H, PhC*H*_2_N), 3.33-3.32 (overlapping, 8H total, 4H, [C3]C*H*_2_N and 4H, [C2]C*H*_2_N), 2.82 (m, 4H, [C2]C*H*_2_N), 2.69 (m, 4H, [C3]C*H*_2_N), 2.02 (m, 4H, CH_2_C*H*_2_CH_2_), 1.98 (overlapping, 12H total, 6H, C*H*_3_COO^−^ and 6H, C*H*_3_COOH). ^1^H NMR (CDCl_3_, 300.1 MHz, 296 K): δ (ppm) 10.61 (overlapping, 6H total, 4H, N*H*_2_^+^ and 2H, COO*H*), 7.54 (d, ^3^*J*_H-H_ = 8 Hz, 2H, *p-Ph*), 7.44 (t, ^3^*J*_H-H_ = 8 Hz, 2H, *m-Ph*), 7.39 (s, 2H, *o-Ph*), 7.36 (d, ^3^*J*_H-H_ = 8 Hz, 2H, *o-Ph*), 3.90 (s, 4H, PhC*H*_2_N), 3.13 (overlapping, 8H total, 4H, [C2]C*H*_2_N and 4H, [C3]C*H*_2_N), 2.77 (m, 4H, [C2]C*H*_2_N), 2.67 (m, 4H, [C3]C*H*_2_N), 2.00 (overlapping, 16H total, 4H, CH_2_C*H*_2_CH_2_, 6H, C*H*_3_COO^−^ and 6H, C*H*_3_COOH). ^13^C{^1^H} NMR (CDCl_3_, 75.5 MHz, 296 K): δ (ppm) 176.7 (*C*OO), 135.5 (*o-Ph*), 134.1 (*i-Ph*), 130.7 (q, ^2^*J*_C-F_ = 32 Hz, *m-Ph*), 128.9 (*m-Ph*), 127.0 (q, ^3^*J*_C-F_ = 4 Hz, *o-Ph*), 124.5 (q, ^3^*J*_C-F_ = 4 Hz, *p-Ph*), 124.2 (q, ^1^*J*_C-F_ = 272 Hz, *C*F_3_), 52.8 (Ph*C*H_2_N), 51.3 ([C3]*C*H_2_N), 48.8 ([C2]*C*H_2_N), 48.5 ([C3]*C*H_2_N or [C2]*C*H_2_N), 46.0 ([C3]*C*H_2_N or [C2]*C*H_2_N), 23.0 (CH_2_*C*H_2_CH_2_), 22.5 (*C*H_3_COO). ^19^F NMR (D_2_O, 376.5 MHz, 296 K): δ (ppm) −62.4 (s, C*F*_3_). ^19^F NMR (CDCl_3_, 282.4 MHz, 296 K): δ (ppm) −62.6 (s, C*F*_3_). Anal. calcd for C_30_H_42_F_6_N_4_O_4_.(CH_3_COOH)_2_: C, 53.96; H, 6.66; N, 7.40. Found: C, 53.85; H, 6.74; N, 7.46.

[H_4_{H_2_(PhCH_2_)_2_Cyclam}]Cl_4_ (**16**): Compound **6** (1.00 g, 2.63 mmol) was dissolved in a minimum volume of ethanol and a concentrated aqueous solution of HCl (37%) was added dropwise until the solution reached pH ≈ 1. The white precipitate which formed was filtered, washed with ethanol, and dried under reduced pressure giving the product in a 51% yield (0.73 g, 1.34 mmol). ^1^H NMR (D_2_O/(CD_3_)_2_CO, 300.1 MHz, 296 K): δ (ppm) 7.57 (overlapping, 10H total, *Ph*), 4.50 (s, 4H, PhC*H*_2_N), 3.68-3.44 (overlapping, 16H total, 8H, [C3]C*H*_2_N and 8H, [C2]C*H*_2_N), 1.19 (m, 4H, CH_2_C*H*_2_CH_2_). ^13^C{^1^H} NMR (D_2_O/(CD_3_)_2_CO, 75.5 MHz, 296 K): δ (ppm) 131.2 (*o-Ph* or *m-Ph*), 129.7 (*p-Ph*), 129.5 (*o-Ph* or *m-Ph*), 129.2 (*i-Ph*), 57.3 (Ph*C*H_2_N), 47.2 ([C2]*C*H_2_N), 44.7 ([C2]*C*H_2_N), 42.1 ([C3]*C*H_2_N), 37.5 ([C3]*C*H_2_N), 17.5 (CH_2_*C*H_2_CH_2_). Anal. calcd for C_24_H_40_Cl_4_N_4_.H_2_O: C, 52.95; H, 7.78; N, 10.29. Found: C, 52.78; H, 7.64; N, 10.25.

[H_4_{H_2_(^4-CF3^PhCH_2_)_2_Cyclam}]Cl_4_ (**17**): Compound **17** was prepared by the same procedure described for **16** using compound **7** (1.00 g, 1.94 mmol) as the starting material. The product was obtained as a white solid in a 66% yield (0.88 g, 1.29 mmol). ^1^H NMR (D_2_O/(CD_3_)_2_CO, 400.1 MHz, 296 K): δ (ppm) 7.85 (d, ^3^*J*_H-H_ = 8 Hz, 4H, *m-Ph*), 7.75 (d, ^3^*J*_H-H_ = 8 Hz, 4H, *o-Ph*), 4.24 (s, 4H, PhC*H*_2_N), 3.65 (m, 4H, [C2]C*H*_2_N), 3.51 (m, 4H, [C3]C*H*_2_N), 3.42 (m, 4H, [C2]C*H*_2_N), 3.20 (m, 4H, [C3]C*H*_2_N), 2.28 (m, 4H, CH_2_C*H*_2_CH_2_). ^13^C{^1^H} NMR (D_2_O/(CD_3_)_2_CO, 100.6 MHz, 296 K): δ (ppm) 136.2 (*i-Ph*), 131.7 (*o-Ph*), 130.8 (q, ^2^*J*_C-F_ = 24 Hz, *p*-Ph), 126.3 (d, ^3^*J*_C-F_ = 3 Hz, *m*-Ph), 124.3 (q, ^1^*J*_C-F_ = 272 Hz, *C*F_3_), 57.1 (Ph*C*H_2_N), 49.8 ([C3]*C*H_2_N), 47.7 ([C2]*C*H_2_N), 44.9 ([C3]*C*H_2_N), 41.2 ([C2]*C*H_2_N), 20.4 (CH_2_*C*H_2_CH_2_). ^19^F NMR (CDCl_3_, 376.5 MHz, 296 K): δ (ppm) −62.6 (s, C*F*_3_). Anal. calcd for C_26_H_38_Cl_4_F_6_N_4_.H_2_O: C, 45.90; H, 5.93; N, 8.23. Found: C, 46.01; H, 5.80; N, 8.23.

[H_4_{H_2_(^4-CH3^PhCH_2_)_2_Cyclam}]Cl_4_ (**18**): Compound **18** was prepared by the same procedure described for **16** using compound **8** (0.80 g, 1.96 mmol) as the starting material. The product was obtained as a white solid in an 88% yield (0.99 g, 1.73 mmol). ^1^H NMR (D_2_O/C_6_D_5_N, 300.1 MHz, 296 K): δ (ppm) 6.51(d, ^3^*J*_H-H_ = 6 Hz, 4H, *o-Ph* or *m-Ph*), 6.51 (d, ^3^*J*_H-H_ = 6 Hz, 4H, *o-Ph* or *m-Ph*), 2.92 (m, 4H, [C3]C*H*_2_N), 2.83 (overlapping, 8H total, 4H, [C2]C*H*_2_N and 4H, PhC*H*_2_N), 2.09 (m, 4H, [C2]C*H*_2_N), 1.96 (m, 4H, [C3]C*H*_2_N), 1.42 (m, 4H, CH_2_C*H*_2_CH_2_) 1.37 (s, 6H, C*H*_3_). ^13^C{^1^H} NMR (D_2_O/C_6_D_5_N, 75.5 MHz, 296 K): δ (ppm) 139.9 (*i-Ph*), 133.2 (*p-Ph*), 132.7 (*o-Ph* or *m-Ph*), 131.6 (*o-Ph* or *m-Ph*), 57.3 ([C3]*C*H_2_N), 53.9 ([C3]*C*H_2_N), 53.0 ([C2]*C*H_2_N), 50.9 (Ph*C*H_2_N), 47.5 ([C2]*C*H_2_N), 24.2 (CH_2_*C*H_2_CH_2_), 22.7 (*C*H_3_). Anal. calcd for C_26_H_44_Cl_4_N_4_.H_2_O: C, 54.55; H, 8.10; N, 9.79. Found: C, 54.41; H, 8.30; N, 9.57.

[H_4_{H_2_(^4-CF3CH2^PhCH_2_)_2_Cyclam}]Cl_4_ (**19**): Compound **19** was prepared by the same procedure described for **16** using compound **9** (0.35 g, 0.64 mmol) as the starting material. The product was quantitatively obtained as a white solid. ^1^H NMR (D_2_O/(CD_3_)_2_CO, 300.1 MHz, 296 K): δ (ppm) 7.63 (d, ^3^*J*_H-H_ = 8 Hz, 4H, *o-Ph* or *m-Ph*), 7.57 (d, ^3^*J*_H-H_ = 8 Hz, 4H, *o-Ph* or *m-Ph*), 4.41 (s, 4H, PhC*H*_2_N), 3.73-3.60 (overlapping, 12H total, 8H, [C2]C*H*_2_N and 4H, ^3^*J*_H-F_ = 11 Hz, C*H*_2_CF_3_), 3.52 (m, 4H, [C3]C*H*_2_N), 3.42 (m, 4H, [C3]C*H*_2_N), 2.33 (m, 4H, CH_2_C*H*_2_CH_2_). ^13^C{^1^H} NMR (D_2_O/(CD_3_)_2_CO, 75.5 MHz, 296 K): δ (ppm) 132.8 (*i-Ph*), 131.6 (*o-Ph* or *m-Ph*), 131.5 (*p-Ph* and *o-Ph* or *m-Ph*), 126.4 (q, ^1^*J*_C-F_ = 277 Hz, CH_2_*C*F_3_), 58.2 (Ph*C*H_2_N), 48.6 ([C3]*C*H_2_N), 45.2 ([C2]*C*H_2_N), 42.4 ([C3]*C*H_2_N), 39.0 (q, ^2^*J*_C-F_ = 29 Hz, *C*H_2_CF_3_), 38.2 ([C2]*C*H_2_N), 19.2 (CH_2_*C*H_2_CH_2_). ^19^F NMR (D_2_O/(CD_3_)_2_CO, 282.4 MHz, 296 K): δ (ppm) −65.6 (s, CH_2_C*F*_3_). Anal. calcd for C_28_H_42_Cl_4_F_6_N_4_.(H_2_O)_2_: C, 46.29; N, 7.71; H, 6.38. Found: C, 46.29; N, 747; H, 6.24.

[H_4_{H_2_(^3-CF3^PhCH_2_)_2_Cyclam}]Cl_4_ (**20**): Compound **20** was prepared by the same procedure described for **16** using compound **10** (0.70 g, 1.36 mmol) as the starting material. The product was obtained as a white solid in a 68% yield (0.61 g, 0.92 mmol). ^1^H NMR (D_2_O/(CD_3_)_2_CO, 300.1 MHz, 296 K): δ (ppm) 7.93-7.92 (overlapping, 8H total, *Ph*), 4.25 (s, 4H, PhC*H*_2_N), 3.76 (m, 4H, [C2]C*H*_2_N), 3.70 (m, 4H, [C3]C*H*_2_N), 3.28 (m, 4H, [C2]C*H*_2_N), 3.09 (m, 4H, [C3]C*H*_2_N), 2.38 (m, 4H, CH_2_C*H*_2_CH_2_). ^13^C{^1^H} NMR (D_2_O/(CD_3_)_2_CO, 75.5 MHz, 296 K): δ (ppm) 128.4 (*i-Ph*), 128.1 (*o-Ph* or *m*-*Ph*), 123.9 (q, ^2^*J*_C-F_ = 32 Hz, *m*-Ph), 123.3 (*o-Ph* or *m*-*Ph*), 121.4 (q, ^1^*J*_C-F_ = 270 Hz, *C*F_3_), 120.4 (q, ^3^*J*_C-F_ = 3 Hz, *o-Ph* or *p*-*Ph*), 118.6 (q, ^3^*J*_C-F_ = 4 Hz, *o-Ph* or *p*-*Ph*), 48.8 (Ph*C*H_2_N), 44.0 ([C3]*C*H_2_N), 42.1 ([C2]*C*H_2_N), 39.7 ([C3]*C*H_2_N), 36.9 ([C2]*C*H_2_N), 15.3 (CH_2_*C*H_2_CH_2_). ^19^F NMR (D_2_O/(CD_3_)_2_CO, 282.4 MHz, 296 K): δ (ppm) -59.8 (s, C*F*_3_). Anal. calcd for C_26_H_38_Cl_4_F_6_N_4_: C, 47.14; H, 5.78; N, 8.46. Found: C, 46.51; H, 5.78; N, 8.40.

[H_4_{H_2_(^4-CF3^PhCH_2_)_2_Cyclam}]Br_4_ (**21**): Compound **7** (0.70 g, 1.36 mmol) was dissolved in a minimum volume of ethanol and a concentrated aqueous solution of HBr (48%) was added dropwise until the solution reached pH ≈ 1. The white precipitate which formed was filtered, washed with ethanol, and dried under reduced pressure giving the product in a 71% yield (0.84 g, 0.96 mmol). ^1^H NMR (D_2_O/(CD_3_)_2_CO, 300.1 MHz, 296 K): δ (ppm) 7.83 (d, ^3^*J*_H-H_ = 8 Hz, 4H, *m-Ph*), 7.72 (d, ^3^*J*_H-H_ = 8 Hz, 4H, *o-Ph*), 4.19 (s, 4H, PhC*H*_2_N), 3.63 (m, 4H, [C3]C*H*_2_N), 3.50 (m, 4H, [C3]C*H*_2_N), 3.37 (m, 4H, [C2]C*H*_2_N), 3.16 (m, 4H, [C2]C*H*_2_N), 2.21 (m, 4H, CH_2_C*H*_2_CH_2_). ^13^C{^1^H} NMR (D_2_O/(CD_3_)_2_CO, 75.5 MHz, 296 K): δ (ppm) 135.4 (*i-Ph*), 131.4 (*o-Ph*), 131.0 (q, ^2^*J*_C-F_ = 33 Hz, *p*-Ph), 126.2 (*m*-Ph), 124.2 (q, ^1^*J*_C-F_ = 273 Hz, *C*F_3_), 57.4 (Ph*C*H_2_N), 49.2 ([C2]*C*H_2_N), 46.7 ([C2]*C*H_2_N or [C3]*C*H_2_N), 43.9 ([C2]*C*H_2_N or [C3]*C*H_2_N), 40.1 ([C3]*C*H_2_N), 19.8 (CH_2_*C*H_2_CH_2_). ^19^F NMR (CDCl_3_, 282.4 MHz, 296 K): δ (ppm) −62.6 (s, C*F*_3_). Anal. calcd for C_26_H_38_Br_4_F_6_N_4_.(H_2_O)_2_: C, 35.64; H, 4.83; N, 6.39. Found: C, 35.73; H, 4.82; N, 6.13.

[H_4_(H_4_Cyclam)]Cl_4_ (**22**): Compound **22** was prepared by the same procedure described for **16** using cyclam (0.59 g, 2.94 mmol) as the starting material. The product was obtained as a white solid in an 89% yield (1.00 g, 2.62 mmol). ^1^H NMR (D_2_O/(CD_3_)_2_CO, 400.1 MHz, 296 K): δ (ppm) 3.59 (m, 8H, [C2]C*H*_2_N), 3.44 (m, 8H, [C3]C*H*_2_N), 2.25 (m, 4H, CH_2_C*H*_2_CH_2_). ^13^C{^1^H} NMR (D_2_O/(CD_3_)_2_CO, 100.6 MHz, 296 K): δ (ppm) 40.5 ([C3]*C*H_2_N), 37.3 ([C2]*C*H_2_N), 17.6 (CH_2_*C*H_2_CH_2_). Anal. calcd for C_10_H_28_Cl_4_N_4_.(H_2_O)_2_: C, 31.43; H, 8.44; N, 14.66. Found: C, 31.34; H, 7.59; N, 14.56.

H_2_(^4-CF3^PhCH_2_)_2_Cyclen (**24**): Compound **23** (0.40 g, 2.06 mmol) was dissolved in the minimum volume of acetonitrile and two equiv. of 4-(trifluoromethyl)benzyl bromide (1.03 g, 4.31 mmol) were added. The solution was stirred overnight at room temperature. The white precipitate which formed was then separated by filtration, washed with acetonitrile, and dried under reduced pressure. The obtained product was heated overnight at 100 ºC in a 5 mL of hydrazine hydrate solution (50%–60%). The mixture was allowed to cool to room temperature and then placed in a cold bath to promote precipitation. The precipitate was filtered off, washed with ethanol, and dried in a vacuum giving compound **24** in a 45% yield (0.45 g, 0.92 mmol). ^1^H NMR (CDCl_3_, 300.1 MHz, 296 K): δ (ppm) 7.60 (d, ^3^*J*_H-H_ = 8 Hz, 4H, *m*-*Ph*), 7.45 (d, ^3^*J*_H-H_ = 8 Hz, 4H, *o*-*Ph*), 3.69 (s, 4H, PhC*H*_2_N), 2.69-2.65 (overlapping, 18H total, 16H, C*H*_2_N and 2H, N*H*). ^13^C{^1^H} NMR (CDCl_3_, 75.5 MHz, 296 K): δ (ppm) 143.4 (*i*-Ph), 129.8 (q, ^2^*J*_C-F_ = 32 Hz, *p*-Ph), 129.3 (*o*-Ph), 125.6 (*q*, ^3^*J*_C-F_ = 4 Hz, *m*-Ph), 124.2 (q, ^1^*J*_C-F_ = 272 Hz, *C*F_3_), 60.1 (Ph*C*H_2_N), 52.1 (*C*H_2_N), 45.7 (*C*H_2_N). ^19^F NMR (CDCl_3_, 282.4 MHz, 296 K): δ (ppm) −62.5 (s, C*F*_3_). Anal. Calcd. for C_24_H_30_F_6_N_4_.½H_2_O: C, 57.94; H, 6.28; N, 11.26. Found: C, 57.70; H, 6.32; N, 11.52.

H_4_[H_2_(^4-CF3^PhCH_2_)_2_Cyclen]Cl_4_ (**25**): Compound **24** (0.43 g, 0.88 mmol) was dissolved in a minimum volume of ethanol and a concentrated aqueous solution of HCl (37%) was added dropwise until the solution reached pH ≈ 1. The brown suspension was centrifugated giving a precipitate that was washed several times with small portions of acetonitrile and diethyl ether and further dried under reduced pressure. Compound **25** was obtained as a yellow solid in a 48% yield (0.27 g, 0.42 mmol). ^1^H NMR (D_2_O/(CD_3_)_2_CO, 300.1 MHz, 296 K): δ (ppm) 7.87 (d, ^3^*J*_H-H_ = 8 Hz, 4H, *m*-*Ph*), 7.75 (d, ^3^*J*_H-H_ = 8 Hz, 4H, *o*-*Ph*), 4.14 (s, 4H, PhC*H*_2_N), 3.50 (m, 8H C*H*_2_N), 3.12 (m, 8H C*H*_2_N). ^13^C{^1^H} NMR (D_2_O/(CD_3_)_2_CO, 75.5 MHz, 296 K): δ (ppm) 140.0 (*i*-Ph), 131.1 (*o*-Ph), 129.8 (q, ^2^*J*_C-F_ = 32 Hz, *p*-Ph), 126.0 (*q*, ^3^*J*_C-F_ = 4 Hz, *m*-Ph), 124.6 (q, ^1^*J*_C-F_ = 271 Hz, *C*F_3_), 56.7 (Ph*C*H_2_N), 47.8 (*C*H_2_N), 42.9 (*C*H_2_N). ^19^F NMR (D_2_O/(CD_3_)_2_CO, 282.4 MHz, 296 K): δ (ppm) −59.8 (s, C*F*_3_). Anal. Calcd. for C_24_H_34_Cl_4_F_6_N_4_: C, 45.44; H, 5.40; N, 8.83. Found: C, 46.04; H, 5.92; N, 10.70.

### 3.3. General Procedures for X-ray Crystallography

Crystallographic and experimental details of data collection and crystal structure determinations for the compounds are available in Appendix A. Suitable crystals of compounds **8** and **14** were coated and selected in Fomblin® oil. Data were collected using graphite monochromated Mo-Kα radiation (λ = 0.71073 Å) on a Bruker AXS-KAPPA APEX II diffractometer. Cell parameters were retrieved using Bruker SMART software and refined using Bruker SAINT on all observed reflections [31]. Absorption corrections were applied using SADABS [32]. The structures were solved by direct methods using SIR97 [33]. Structure refinement was done using SHELXL [34], included in the WINGX-Version 1.80.01 [35] system of programs. The hydrogen atoms of the N*H* and COO*H* groups were located in the electron density maps. The other hydrogen atoms were inserted in calculated positions and allowed to refine in the parent atoms. Torsion angles, mean square planes, and other geometrical parameters were calculated using SHELX [34]. Illustrations of the molecular structures were made with Mercury CSD 3.9 for Windows [36]. Data for the structures of compounds **8** and **14** were deposited in CCDC under the deposit numbers 1920400 and 1920401, respectively, and can be obtained free of charge from The Cambridge Crystallographic Data Centre via http://www.ccdc.cam.ac.uk/conts/retrieving.html.

### 3.4. Minimal Inhibitory Concentration Assays

*Escherichia coli* ATCC 25922 and *Staphylococcus aureus* Newman are human clinical isolates and were maintained in Lennox broth (LB) solid medium. Minimal inhibitory concentration (MIC) assays were performed in Mueller–Hinton broth (MHB; Becton, Dickinson and Company) using a microdilution assay, based on previously described methods [21,37]. Briefly, bacterial cultures freshly grown in MBH solid medium contained in Petri plates were transferred into MHB liquid medium and grown for 4–5 hours with orbital agitation (250 rpm) at 37 °C. The cultures were then adequately diluted in fresh MHB to obtain approximately 10^6^ colony forming units (CFUs) per mL. Adequate volumes of these cultures were used to inoculate approximately 5 × 10^5^ CFUs per mL in 96-well polystyrene microtiter plates containing 100 μL of MHB supplemented with different concentrations of each compound under study, achieved by 1:2 serial dilutions ranging 512 μg/mL to 0.5 μg/mL. Compounds were prepared with distilled water and filtered with a 0.22 μm sterile filter. As the positive control, aliquots of 100 μL of 1× concentrated MHB and 100 μL containing 10^6^ CFUs per mL were used, while for negative controls, aliquots of 200 μL of sterile MHB were used. The compounds sterility was also tested. The microtiter plates were then incubated at 37 °C for 20 h and bacterial growth was assessed by determining the optical density (OD) of cultures at 640 nm using a SPECTROstarNano (BMG LABTECH) microplate reader. Experiments were carried out at least four times. 

## 4. Conclusions

In summary, our previous studies attested to the antibacterial properties of cyclam-based compounds. The results described herein allow for the establishment of the precise relations between the antibacterial activity and the structural features of the compounds derived from saturated tetraazamacrocycles. Specifically, it has been shown that: i) the number of pendant arms on the macrocyclic backbone; ii) the nature; iii) the relative positions of the substituents on the aromatic ring of the pendant arms; and iv) the size of the macrocycle cavity, are critical issues. The design of new macrocyclic molecules that systematically display variations of these properties, aiming to highlight and optimize their antimicrobial activity is a current topic of much interest to us.

The beneficial effect of the coordination of Fe^2+^ and Mn^2+^ to tetraazamacrocycles has been reported by Hubin et al. [14,16] and Khan et al. [15]. Our future research will investigate the antimicrobial activities of metal complexes supported by cyclam and cyclen ligands.

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
