# Peer review of "Investigations into the Structure/Antibacterial Activity Relationships of Cyclam and Cyclen Derivatives"

_antibiotics, 2019, doi:10.3390/antibiotics8040224_

Round 1

Reviewer 1 Report

Alves and co-authors reported a series of cyclam and cyclen-derived salts, designed to gain insights into their structure and antibacterial activity towards Staphylococcus aureus and Escherichia coli. The results obtained show that the chemical nature, polarity and substitution patterns of the benzyl groups, as well as the number of pendant arms are critical parameters for the antibacterial activity of the cyclam-based salts.

The reviewer believes the manuscript was well written, the science behind is sound and the discussion was solid, this work will attract great attentions from the scientific community of medicinal chemistry, organic synthesis and others. Therefore, the reviewer recommends the acceptance of this paper after addressing a few suggestions below:

Could the author explain why there is a slight reduction of the antibacterial activity of cyclam derivatives when insert one CH2 spacer between the trifluoromethyl group and the aromatic ring? (pairs 12/14 and 17/19) Follow up with the first question, the author investigated the effect of the distance of the trifluoromethyl group attached to the aromatic ring of the macrocyclic pendant arm on the antibacterial activity of the compound. Did author test longer spacer compounds and how about the activities?

Author Response

Rev #1: Alves and co-authors reported a series of cyclam and cyclen-derived salts, designed to gain insights into their structure and antibacterial activity towards Staphylococcus aureus and Escherichia coli. The results obtained show that the chemical nature, polarity and substitution patterns of the benzyl groups, as well as the number of pendant arms are critical parameters for the antibacterial activity of the cyclam-based salts.

The reviewer believes the manuscript was well written, the science behind is sound and the discussion was solid, this work will attract great attentions from the scientific community of medicinal chemistry, organic synthesis and others. Therefore, the reviewer recommends the acceptance of this paper after addressing a few suggestions below:

Could the author explain why there is a slight reduction of the antibacterial activity of cyclam derivatives when insert one CH2 spacer between the trifluoromethyl group and the aromatic ring? (pairs 12/14 and 17/19) Follow up with the first question, the author investigated the effect of the distance of the trifluoromethyl group attached to the aromatic ring of the macrocyclic pendant arm on the antibacterial activity of the compound. Did author test longer spacer compounds and how about the activities? 

Answer: The authors gratefully acknowledge the general appreciation and comments of the reviewer. Concerning the question related with the slight reduction of antibacterial activity when a CH2 spacer was inserted, we haven´t synthesized or tested compounds with longer spacers. However, we also think this is an interesting issue and worth to further explore. Although no additional work was done, we hypothesize that this reduction is related with the interruption of the electron delocalization between the CF3 moiety and the aromatic ring, increasing the local polarity on the pendant arm. Furthermore, the introduction of the CH2 spacer also introduce geometrical changes, with the CF3 moiety changing from a linear position to a bend position related to the aromatic ring. In order to give the readers our interpretation of this reduction in activity due to the spacer, we added the following sentences to the revised version, new lines 223-227, which reads as follows:

“The decrease of antimicrobial activity may have either electronic or stereochemical origin. The inclusion of a CH2 spacer has a strong influence on the relative position of the CF3 group face to a possible acceptor fragment and thus, it is expected to modify the interaction between both fragments. Additionally, the CH2 spacer blocks the electronic delocalization that is present if the CF3 group is directly bonded to the aromatic ring.)”

Reviewer 2 Report

A brief summary (one short paragraph) outlining the aim of the paper and its main contributions.

This manuscript outlines the simple syntheses of a number of cyclam and cyclen derivatives bearing one or two trans pendant arms on the macrocyclic nitrogens.  All pendants are benzyl based, with some benzyl substitution, primarily with methyl or trifluoromethyl groups.  The syntheses follow standard literature procedures.  Once made, different salts (acetate, chloride, and bromide) salts of the macrocycles are made.  The compounds are then tested for MIC against S. aureus and E. coli as models for Gram-positive and Gram-negative bacteria.  Structure/activity relationships are presented based on macrocycle ring size, number and type of pendant arm, and anion.  None of the compounds appear to be strong candidates as drug leads.

Broad comments highlighting areas of strength and weakness. These comments should be specific enough for authors to be able to respond.

The synthetic work is done expertly and presented with careful characterization.  Similarly, the data for the antimicrobial studies is carried out and presented in a way that leads to high confidence in the results.  One weakness is the small number of compounds tested, so that that structure/activity relationships are tenuous, especially regarding the singular mono-substituted compound, and the singular cyclen derivative.  One compound seems not enough to draw very strong conclusions about.  An additional weakness is lack of awareness of the literature, which contains very similar compounds that have already been tested for antimicrobial (antifungal, antischistosomal, antileishmanial, and antibiotic) activity.  Comparisons to some of these compounds structures and activities would be quite helpful, in fleshing out the possible structure/activity relationships that are the aim of this manuscript.

Specific comments referring to line numbers, tables or figures. Reviewers need not comment on formatting issues that do not obscure the meaning of the paper, as these will be addressed by editors.

This reviewer is aware of a number papers from the literature that should be added to the introduction and likely to the structure/activity discussion.  Specifically, the following papers should be examined by the authors and cyclam/cyclen mono- and bis- disubstituted compounds with antimicrobial activities already known should be incorporated into the discussion:

“Tetraazamacrocyclic derivatives and their metal complexes as antileishmanial leads”

Timothy J. Hubin, Ashlie N. Walker, Dustin J. Davilla, TaRynn N. Carder Freeman, Brittany M. Epley, Travis R. Hasley, Prince N.A. Amoyaw, Surendra Jain, Stephen J. Archibald, Timothy J. Prior, Jeanette A. Krause, Allen G. Oliver, Babu L. Tekwani, M. Omar F. Khan  Polyhedron 163 (2019) 42–53

“Discovery of Antischistosomal Drug Leads Based on Tetraazamacrocyclic Derivatives and Their Metal Complexes” M. O. Faruk Khan, Jennifer Keiser, P. N. A. Amoyaw, Mohammad F. Hossain, Mireille Vargas, Justin G. Le, Natalie C. Simpson, Kimberly D. Roewe, TaRynn N. Carder Freeman, Travis R. Hasley, Randall D. Maples, Stephen J. Archibald, Timothy J. Hubin  Antimicrobial Agents and Chemotherapy, 2016, 60, 5331-5336.

Author Response

Reviewer # 2: This manuscript outlines the simple syntheses of a number of cyclam and cyclen derivatives bearing one or two trans pendant arms on the macrocyclic nitrogens. All pendants are benzyl based, with some benzyl substitution, primarily with methyl or trifluoromethyl groups.  The syntheses follow standard literature procedures. Once made, different salts (acetate, chloride, and bromide) salts of the macrocycles are made. The compounds are then tested for MIC against S. aureus and E. coli as models for Gram-positive and Gram-negative bacteria. Structure/activity relationships are presented based on macrocycle ring size, number and type of pendant arm, and anion. None of the compounds appear to be strong candidates as drug leads.

The synthetic work is done expertly and presented with careful characterization. Similarly, the data for the antimicrobial studies is carried out and presented in a way that leads to high confidence in the results. One weakness is the small number of compounds tested, so that structure/activity relationships are tenuous, especially regarding the singular mono-substituted compound, and the singular cyclen derivative. One compound seems not enough to draw very strong conclusions about. An additional weakness is lack of awareness of the literature, which contains very similar compounds that have already been tested for antimicrobial (antifungal, antischistosomal, antileishmanial, and antibiotic) activity. Comparisons to some of these compounds structures and activities would be quite helpful, in fleshing out the possible structure/activity relationships that are the aim of this manuscript.

Answer: The authors gratefully acknowledge the comments of this reviewer, which we have thoroughly used to improve the manuscript. In this work, a total of 14 compounds were studied, some of them synthesized for the first time. Although we agree this is a limited number of compounds, we still believe that some conclusions can be taken. In order to accommodate this criticism, some words were eliminated to tone down some of our conclusions, e.g. “dramatic effect” was substituted by “effect” on new line 215.

Concerning the macrocyclic backbone size, we agree that conclusions are limited since we are comparing only 2 compounds, the disubstituted cyclen and cyclam. Although we haven´t referred in the text we have successfully synthesized a tetrasubstituted cyclam of formula (4-CF3PHCH2)4Cyclam. This compound and its chloride salt are not soluble in water, which hampered the evaluation of their antimicrobial activity. Due to this behavior we opted not to present its synthesis in the manuscript. To accommodate the criticisms, we have rephrased the sentence which now reads in the revised version as follows in new lines 235-238: “This result suggests that the size of the macrocyclic backbone is an important feature for the antimicrobial activity of the tetraazamacrocycles, although strong conclusions cannot be taken on this subject due to the limited number of compounds tested.”

Concerning the effects of the counter anion on the antimicrobial activity we believe that we have enough compounds to take solid conclusions. Therefore, we rephrased the original sentence which now reads as follows in the new lines 212-214: Such observations are more pronounced for S. aureus than for E. coli, and might reflect the distinct and specific mechanisms used by each bacterial species to cope with antimicrobials.

An additional comment from the reviewer concerns the reduced references to the various biological activities of tetraazamacrocycles derivatives and metal complexes. We recognize that our initial version was lacking some important work already published. Therefore, we added some information to the introduction part to surpass this issue. Please see new lines 60-65, which reads as follows: “Tetraazamacrocycles and their metal complexes have also been studied as antimalarial [13,14], antischistosomal [15] as well as antileishmanial drugs [16]. Furthermore, the latter compounds have been investigated as anticancer agents [17]. In this context, a Zn2+ tetraazamacrocycle complex exhibiting a considerable cytotoxicity towards human breast, cervical and lung cancer cell lines was recently described [18]. The interaction with DNA suggests that the mechanism of apoptotic induction may be distinct of that of cisplatin. [18]”

Reviewer #2: This reviewer is aware of a number papers from the literature that should be added to the introduction and likely to the structure/activity discussion. Specifically, the following papers should be examined by the authors and cyclam/cyclen mono- and bis- disubstituted compounds with antimicrobial activities already known should be incorporated into the discussion:

“Tetraazamacrocyclic derivatives and their metal complexes as antileishmanial leads”

Timothy J. Hubin, Ashlie N. Walker, Dustin J. Davilla, TaRynn N. Carder Freeman, Brittany M. Epley, Travis R. Hasley, Prince N.A. Amoyaw, Surendra Jain, Stephen J. Archibald, Timothy J. Prior, Jeanette A. Krause, Allen G. Oliver, Babu L. Tekwani, M. Omar F. Khan. Polyhedron 163 (2019) 42–53

 “Discovery of Antischistosomal Drug Leads Based on Tetraazamacrocyclic Derivatives and Their Metal Complexes” M. O. Faruk Khan, Jennifer Keiser, P. N. A. Amoyaw, Mohammad F. Hossain, Mireille Vargas, Justin G. Le, Natalie C. Simpson, Kimberly D. Roewe, TaRynn N. Carder Freeman, Travis R. Hasley, Randall D. Maples, Stephen J. Archibald, Timothy J. Hubin. Antimicrobial Agents and Chemotherapy, 2016, 60, 5331-5336.

Answer: We thank the suggestion of the reviewer. As can be seen from the previous answer, these two references were added to the “Introduction” section, together with other references, Please see new lines 60-65 in the introduction part. In addition, in the “Conclusions” section we have also incorporated the mentioned references, please see new lines 365-367, which now reads as follows: “The beneficial effect of the coordination of Fe2+ and Mn2+ to tetraazamacrocycles has been reported by Hubin et al. [14, 16] and Khan et al. [15]. Further developments of our work will investigate the antimicrobial activities of metal complexes supported by cyclam and cyclen ligands.”

Round 2

Reviewer 2 Report

The authors have adequately addressed my concerns with the first draft.  I would support publication in its present form.